# The Association between Dietary Magnesium Intake and Magnetic Resonance Parameters for Knee Osteoarthritis

**DOI:** 10.3390/nu11061387

**Published:** 2019-06-20

**Authors:** Nicola Veronese, Luciana La Tegola, Maria Gabriella Caruso, Stefania Maggi, Giuseppe Guglielmi

**Affiliations:** 1National Research Council, Neuroscience Institute, Aging Branch, 35128 Padova, Italy; stefania.maggi@in.cnr.it; 2Research Hospital, National Institute of Gastroenterology, IRCCS De Bellis, BA 70077 Castellana Grotte, Italy; gabriella.caruso@irccsdebellis.it; 3Università degli Studi di Foggia, Scuola di Specializzazione di Area Medica, Department of Radiology, 71100 Foggia, Italy; luciana.lategola@hotmail.it (L.L.T.); giuseppe.guglielmi@unifg.it (G.G.)

**Keywords:** knee osteoarthritis, magnesium, MRI, aged, healthy ageing, lifestyle

## Abstract

The aim of the study was to evaluate the relationship between dietary magnesium (Mg) intake and prevalence of knee osteoarthritis (OA), a topic poorly explored in the literature. Overall, 783 people participating in the Osteoarthritis Initiative (59.8% females; mean age: 62.3 years) and having an MRI assessment were enrolled in this cross-sectional study. Mg intake was measured with a semi-quantitative food frequency questionnaire, and its association with knee OA was evaluated for an increased intake of 100 mg/day. Using an adjusted linear regression analysis, a higher Mg intake (i.e., increase of 100 mg/day) corresponded to a significant increase in mean cartilage thickness, cartilage volume at medial tibia, cartilage volume and mean cartilage thickness at central medial femur, and cartilage volume and mean cartilage thickness in the central medial tibiofemoral compartment. In conclusion, an increased Mg dietary intake is associated with a better knee cartilage architecture, also when adjusting for potential confounders, suggesting a potential role of Mg in the prevention and treatment of knee OA.

## 1. Introduction

One of the major causes of disability in aged people is osteoarthritis (OA). Inflammation plays a central role in OA [1], but the etiology of OA is multi-factorial and mainly unknown. We know several risk factors for knee OA including age, gender, ethnicity, genetics. Among them, age is probably one of the strongest predictors of OA. Currently, the global prevalence of OA is estimated as 10% in men and 20% in women over 60 [2] but it is increasing because of the growth of the elderly population. Therefore, understanding the mechanisms of OA is important, in order to reduce its prevalence.

Several minerals such as calcium, magnesium (Mg), sodium, potassium, selenium, zinc, copper, and iron might regulate cartilage normal function [3]. Particularly, in musculoskeletal disorders, Mg has been overlooked for a long time, until recent years [4,5]. Approximately half of the US population consumes less than the daily requirement of Mg from foods and water sources [6], probably due to the consumption of processed foods and filtered water rather than ground water which limits Mg assumption.

Several epidemiological studies (as summarized in [7]) suggested an association among dietary Mg deficiency and different diseases, such as atherosclerosis [8], hypertension [9], osteoporosis [10], diabetes mellitus [11], colon and breast cancers [12]. Inflammatory responses seem to underlie the relationship between low Mg and these conditions. [13] Since inflammation can play a central role in OA [1], it has been hypothesized that Mg intake would be inversely associated with knee OA. Animal studies, in fact, reported that Mg deficiency induces a relevant inflammatory response finally resulting in leukocyte and macrophage activation, increase in inflammatory cytokines and acute-phase proteins, and excessive production of free radicals [13].

Important for the study of OA is magnetic resonance imaging (MRI) because it permits the visualization of structures undetectable on plain radiograph (e.g., articular cartilage, meniscus, ligaments, synovial, capsular structures, fluid collection, and bone marrow) and the evaluation of pre-radiographic changes.

Given the potential role of Mg in different diseases and the limited literature regarding Mg and knee OA parameters analyzed through MRI, we therefore assessed the potential association between Mg intake and knee OA architecture.

## 2. Materials and Methods

### 2.1. Data Source and Subjects

These data were collected from the Osteoarthritis Initiative (OAI) database (http://www.oai.ucsf.edu/). This database contains information of participants living in four cities in the United States (Baltimore, MD; Pittsburgh, PA; Pawtucket, RI; and Columbus, OH) enrolled between February 2004 and May 2006 [14].

After being informed of the aims and methods of the study, the participants provided informed written consent. The OAI study obtained full ethical approval by the institutional review board of the OAI Coordinating Center at University of California, San Francisco.

### 2.2. Dietary Magnesium Intake (Exposure)

Dietary Mg intake was obtained through a food frequency questionnaire recorded at the OAI baseline assessment, including Mg supplementation [10,15,16]. Then, the population was categorized in quartiles using 190, 265, 320 mg/day as cut-offs (for descriptive purposes) and an increase of 100 mg/day of Mg intake as an independent variable (in the fully adjusted analyses).

### 2.3. Outcomes

Coronal 3D FLASH images with Water Excitation MR sequence of the osteoarthritic right knee acquired at 3 T (Siemens Magnetom Trio, Erlangen, Germany) were analyzed. It is a double oblique 3D fast low-angle shot (FLASH) with water excitation, a slice thickness of 1.5 mm, and an in-plane resolution of 0.31 mm × 0.31 mm [17].

In pairs, seven blinded and experienced technicians read the images [18]. The total subchondral bone area and surface area of the cartilage joint (AC) were calculated in the medial (MT) and in the lateral (LT) tibial compartment and in the central (weight-bearing) medial (cMF) and central lateral (cLF) femoral condyle. The weight-bearing region of the femoral condyles was analyzed between the intercondylar notch and 60% of the distance to the posterior end of the femoral condyles used as the anatomical landmark. The medial (MFTC) and lateral (LFTC) femur-tibial compartments for total cartilage plates and for the central sub-regions were also reported, by using summed values from MT and cMF, and LT and cLF, respectively. All segmentations were quality controlled by a single expert (S.M.) [18].

The total subchondral bone area, the AC, the part of the subchondral bone covered with cartilage (cAB), the denuded subchondral bone area (dAB), the cartilage volume (VC), and the mean cartilage thickness (ThCcAB over cAB and ThCtAB over AC) were then computed [18]. Additional details on these procedures are available from the OAI imaging protocol [19].

Since the cartilage thickness and volume appear to be the best predictors of the bone–cartilage interface in the knee [18], the current study was particularly interested in the ThCtAB and the VC of the MT and LT, in cLF and cMF and in LFTC and MFTC.

### 2.4. Covariates

We identified several potential confounders in our analyses: body mass index (BMI), physical activity level, assessed with the Physical Activity Scale for the Elderly (PASE) [20], race, smoking habit, educational level, yearly income (< or >$50,000 and missing data), self-reported comorbidities, assessed using the modified Charlson comorbidity score [21], use of analgesic drugs (both topical and systemic) for the management of pain-related knee OA, number of alcoholic drinks consumed in a typical week in the previous 12 months, and adherence to the Mediterranean diet (aMED) [22,23,24,25].

### 2.5. Statistical Analyses

Continuous variable data were normally distributed according to the Kolmogorov–Smirnov test. Data are presented as means and standard deviation (SD) values for quantitative measurements, frequency and percentages for discrete variables. To test the homoschedatic variance, the Levene’s test was used; if the assumption was violated, Welch’s ANOVA was used. The *p* values were calculated using the Jonckheere–Terpstra test [25] for continuous variables and the Mantel–Haenszel Chi-square test for the categorical ones.

Using the MRI parameters of the knee as “outcome” and Mg as “exposure”, a linear regression analysis was performed to evaluate their relationship. The basic model was not adjusted for any confounding factors, while the fully adjusted model included the covariates cited in paragraph 2.4. The covariates were selected among those factors significantly associated with at least three results in univariate analyzes, using a *p* value <0.10 for inclusion. The multi-collinearity was evaluated through the variance inflation factor (VIF) [26], using a cut-off of 2 as reason for exclusion, and, adopting this criterion, alcohol intake was removed since highly collinear with aMED (VIF = 2.58). Adjusted standardized betas with their 95% confidence intervals (CIs) were calculated to estimate the strength of the associations between Mg (increase of 100 mg/day) and knee MRI parameters.

Several sensitivity analyses were conducted with SPSS software version 21.0 for Windows (SPSS Inc., Chicago, IL, USA) to evaluate the interaction between dietary Mg intake and selected factors (such as median age, sex, presence of comorbidities, presence of clinical/radiological knee OA, median BMI, education, income, smoking status, race, use of Mg supplementation) in the association with knee MRI parameters, but none emerged as a moderator of our findings (all *p*-values >0.05 for the interaction). A *p* < 0.05 was considered statistically significant.

## 3. Results

### 3.1. Sample Selection

The OAI study initially included 4796 North American participants. Of them, 782 were included in our research.

### 3.2. Descriptive Characteristics

In total, 468 of the 782 participants (59.8%) were females, with an average age of 62.3 years (±9.4 years, range: 45–79). The average intake of Mg in the participants’ diet (diet and supplementation) was 266 ± 112 (range: 57–955) mg/day, with only 137 (17.5%) participants reaching the corresponding RDA.

Table 1 illustrates the baseline characteristics of the subjects classified on the basis of the quartiles of intake of dietary Mg. Those in the highest quartile were more likely to be female (*p* for the trend = 0.02) and wealthier (*p* for the trend = 0.004) than the other participants. The participants who ingested more Mg with their diet had a significantly higher calorie intake and adherence to the Mediterranean diet (*p* for the trend <0.0001 for both parameters).

### 3.3. Dietary Magnesium Intake and Knee MRI Parameters

Table 2 shows the association between dietary Mg intake and knee MRI parameters. Using a linear regression analysis, adjusted for 11 potential confounding factors, a higher Mg intake (i.e., a 100 mg/day increase with diet and/or supplementation) corresponded to a significant increase in mean cartilage thickness at the medial tibia (beta = 0.02; 95%CI: 0.01–0.04; *p* = 0.049), cartilage volume at the medial tibia (beta = 0.08; 95% CI: 0.04–0.12; *p* = 0.02), cartilage volume at the central medial femur (beta = 0.08; 95% CI: 0.04–0.13; *p* = 0.03), mean thickness of the cartilage at the femur central medial (beta = 0.10; 95% CI: 0.06–0.13; *p* = 0.046), mean cartilage thickness at the central medial tibial-femoral compartment (beta = 0.10; 95% CI: from 0.05–0.15; *p* = 0.03), and cartilage volume at the same site (beta = 0.08; 95% CI: 0.04–0.12; *p* = 0.04).

Sensitivity analysis (i.e., stratification by median age, sex, presence of any comorbidities, presence/absence of OA of the radiological/clinical knee, median BMI, education, income, smoking status, race, use of Mg supplementation) showed no significant interaction between dietary Mg intake and these factors in the association of Mg intake with knee MRI parameters (details not shown, available upon request).

## 4. Discussion

In this cross-sectional investigation, involving a large cohort of North Americans at risk of knee osteoarthritis or reporting this condition, an increased dietary Mg intake was associated with better knee morphology, even independently of several confounding factors. In participants with an augmented Mg intake, we reported a significant increase in the volume and thickness of cartilage in the central region of the medial femur and in the central medial tibiofemoral compartment.

These results add some novel findings to the previous literature reporting a possible association between dietary and serum Mg intake with prevalent radiographic knee OA [27,28,29] and overall suggesting that Mg may have a protective effect on knee OA. In fact, as some studies have already shown, the thinning and loss of cartilage are the first features of OA and can be detected only by magnetic resonance imaging [30,31]. An optimal baseline cartilage thickness and volume are, consequently, important factors in the onset of knee OA [30]. In this regard, to be overweight or obese can contribute to knee OA by increasing pressure on lower limbs joints. Excessive weight, in fact, can increase the stress on joints and finally lead to cartilage damage [26]. Therefore, an increased dietary intake of Mg could delay the onset or attenuate the initial phase of OA in people at higher risk, also because Mg intake seems to be associated with a better body composition in older people [7].

We can put forward some hypotheses to explain our findings. First, low-grade inflammation and pathological conditions for which inflammatory stress is considered a risk factor are often associated with Mg deficiency [13,32,33]. As reported in the Introduction, inflammation may play an important role in the destruction of knee cartilage in osteoarthritis [34]. Furthermore, animal studies show that Mg deficiency intensifies the recruitment of phagocytes with the generation of reactive oxygen species (ROS) [35]. OA progression is significantly related to oxidative stress and ROS [36], suggesting that adequate intake and/or supplementation of Mg can reduce inflammation and oxidative stress, leading to better preservation of knee structures. Third, Mg is particularly present in the Mediterranean diet, rich in vegetables, and, consequently, this can further justify our findings [22,37,38]. Finally, it is widely known that Mg is able to potentiate the effects of vitamin D on the skeletal system, further reinforcing the positive effect of Mg on knee OA [27].

Our results are also supported by studies carried out in animals reporting that the deficiency of Mg can be associated with an apparent decrease in the number and size of the proximal tibial articular chondrocytes, with a consequent decrease in the width of the articular cartilage and a significant decrease in the trabecular bone volume [39]. In human beings, as mentioned before, numerous epidemiological studies have reported that the Mg is inversely associated with the radiographic knee OA [29,30].

Therefore, our study using MRI parameters is in agreement with the literature regarding this topic and reporting the association between Mg and knee OA prevalence [28,29,30].

The results of our study should be interpreted considering its shortcomings. The main limitation of this study is its cross-sectional nature that precluded any possibility of examining a potential causal relationship between Mg deficiency and knee MRI parameters. Secondly, it did not examine bio-humoral markers and other serum minerals and their effects on each other and on the association between Mg and knee status, which could actually be relevant. In this regard, however, we would like to remember that plasma and serum Mg, commonly used for assessing Mg status, are not good estimators of Mg levels in human beings [31]. Finally, physical performance was assessed through the PASE that seems to be poorly associated with other nutritional and metabolic parameters such as the SCREEN score [32]. On the other hand, among the strengths of our study, we can list the large sample size and the fact that this is the first epidemiological study that reports data on the impact of Mg deficiency on knee status evaluated by magnetic resonance.

## 5. Conclusions

To conclude, our findings confirm that a higher dietary intake of Mg is associated with a significant increase in MRI-proven knee joint cartilage and volume, which has an important role in preventing OA, even after taking in consideration several important confounding factors. Further longitudinal studies are warranted to confirm or refute these results and explore potential pathophysiological mechanisms underlying these findings.

## Figures and Tables

**Table 1 nutrients-11-01387-t001:** Descriptive findings of the participants by total dietary magnesium intake.

	Q1 (*n* = 196) Mg < 190 mg/day	Q2 (*n* = 195) Mg 190–264 mg/day	Q3 (*n* = 196) Mg 265–319 mg/day	Q4 (*n* = 195) Mg > 320 mg/day	*p* Value ^a^
General characteristics					
Energy intake (Kcal/day)	1076 (223)	1247 (285)	1564 (413)	1940 (584)	<0.0001
aMED (points)	26.5 (4.69	28.3 (4.9)	28.9 (4.6)	29.5 (4.7)	<0.0001
Alcoholic drinks in a typical week (n)	1.59 (1.29)	1.74 (1.36)	1.84 (1.66)	1.95 (1.63)	0.10
Age (years)	61.8 (9.4)	62.3 (9.4)	62.2 (9.5)	62.8 (9.5)	0.34
PASE (points)	149 (78)	164 (78)	158 (77)	154 (76)	0.26
Females (n, %)	65.8	59.5	60.2	53.3	0.02
White race (n, %)	83.2	84.6	88.3	84.6	0.48
Smoking (previous/current)	44.8	47.9	46.2	46.1	0.90
Graduate degree (n, %)	24.5	26.2	30.6	30.8	0.11
Yearly income (< 50,000 $)	52.0	55.9	63.8	63.1	0.004
Medical conditions					
BMI (Kg/m^2^)	29.5 (4.7)	29.5 (4.9)	29.3 (4.8)	29.2 (4.4)	0.37
Any pain killer (%)	41.3	47.2	44.9	46.4	0.42
Charlson co-morbidity index (points)	0.53 (1.00)	0.44 (0.94)	0.40 (0.72)	0.36 (0.71)	0.36

**Notes:** The data are presented as means (with standard deviations) for continuous variables and number (with percentage). **^a^**
*p* values for trends were calculated using the Jonckheere–Terpstra test for continuous variables and the Mantel–Haenszel Chi-square test for categorical ones. Abbreviations: aMED: adherence to Mediterranean diet; PASE: Physical Activity Scale for the Elderly; BMI: body mass index.

**Table 2 nutrients-11-01387-t002:** Association between dietary magnesium intake and magnetic resonance parameters.

	Unadjusted Linear Regression			Fully Adjusted ^1^ Linear Regression		
	Beta	95%CI beta	*p*-value	Beta	95%CI beta	*p* value
Mean cartilage thickness—medial tibia	0.09	0.05 to 0.13	0.01	**0.02**	**0.01 to 0.04**	**0.049**
Volume of cartilage—medial tibia	0.11	0.08 to 0.15	0.003	**0.08**	**0.04 to 0.12**	**0.02**
Volume of cartilage—lateral tibia	0.08	0.04 to 0.11	0.04	0.02	−0.02 to 0.06	0.57
Mean cartilage thickness—lateral tibia	0.03	−0.01 to 0.06	0.49	0.01	−0.03 to 0.06	0.81
Volume of cartilage—central lateral femur	0.09	0.05 to 0.13	0.02	0.04	−0.01 to 0.08	0.45
Mean cartilage thickness—central lateral femur	0.06	0.02 to 0.10	0.01	0.04	−0.01 to 0.08	0.45
Volume of cartilage—central medial femur	0.10	0.06 to 0.14	0.007	**0.08**	**0.04 to 0.13**	**0.03**
Mean cartilage thickness—central medial femur	0.10	0.06 to 0.13	0.009	**0.10**	**0.06 to 0.13**	**0.046**
Mean cartilage thickness—lateral tibial-femoral compartment	0.05	−0.009 to 0.08	0.22	0.02	−0.02 to 0.07	0.59
Cartilage volume—medial tibial-femoral compartment	0.09	0.05 to 0.13	0.02	0.03	−0.008 to 0.06	0.43
Mean cartilage thickness—central medial tibial-femoral compartment	0.10	0.08 to 0.15	0.002	**0.10**	**0.05 to 0.15**	**0.03**
Cartilage volume—medial tibial-femoral compartment	0.11	0.08 to 0.15	0.002	**0.08**	**0.04 to 0.12**	**0.04**

**Notes:** Data are presented as standardized beta with 95% confidence intervals and corresponding *p*-values and were obtained through a linear regression analysis taking the increase of 100 mg of dietary magnesium intake (from both supplementation and diet) as an independent variable. Significant findings (*p* < 0.05), after full adjustment, are reported in bold. **^1^** The fully adjusted model included as covariates: age (as continuous), sex, race (whites vs others), body mass index (as continuous), education (degree vs others,; smoking habits (current and previous vs others). yearly income (categorized as > or <$50,000 and missing data), Charlson co-morbidity index, physical activity scale for elderly (as continuous), total energy intake (as continuous), adherence to the Mediterranean diet.

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
