# Peer review of "The Association between Dietary Magnesium Intake and Magnetic Resonance Parameters for Knee Osteoarthritis"

_nutrients, 2019, doi:10.3390/nu11061387_

Round 1
Reviewer 1 Report
In this manuscript, the authors showed that higher dietary magnesium intake was associated with a better knee cartilage architecture & suggested a potential beneficial role of Mg in the prevention & treatment of knee Osteoarthritis.
Authors should discuss how higher magnesium might promote bone health. One of the potential mechanisms could be related to potentiating the effects of vitamin D functions on skeletal system; as magnesium is essential for vitamin D activation & function (J Am Osteopath Assoc. 2018 Mar 1;118(3):181-189. doi: 10.7556/jaoa.2018.037).
Authors also need to let the readers know that serum magnesium level does not always reflect the total magnesium status of the body (Nutrients. 2018 Dec 2;10(12). pii: E1863. doi: 10.3390/nu10121863).
The main finding of the study is already known, & therefore, the newness of this manuscript is not obvious, except that MRI was used.
This manuscript needs extensive language editing. There are typos & unusual contractions of sentences.
Author Response
Reviewer 1:
Authors should discuss how higher magnesium might promote bone health. One of the potential mechanisms could be related to potentiating the effects of vitamin D functions on skeletal system; as magnesium is essential for vitamin D activation & function (J Am Osteopath Assoc. 2018 Mar 1;118(3):181-189. doi: 10.7556/jaoa.2018.037).
R: We have added this concept and the reference in the Discussion section.
Authors also need to let the readers know that serum magnesium level does not always reflect the total magnesium status of the body (Nutrients. 2018 Dec 2;10(12). pii: E1863. doi: 10.3390/nu10121863).
R: We sincerely thank the Reviewer for this comment. We have now added in the Limitations section this important topic.
The main finding of the study is already known, & therefore, the newness of this manuscript is not obvious, except that MRI was used.
R: We have added a sentence regarding this point. However, we would like to note that the literature regarding Mg and knee OA is limited to a few studies and we are the first in using the MRI for assessing knee structure.
This manuscript needs extensive language editing. There are typos & unusual contractions of sentences.
R: The manuscript was now revised by an English native speaker.
Reviewer 2 Report
To enrich the introduction I suggest the authors to include an explanation for the link of Mg deficiency and inflammation processes. Hence, the readers may ascertain whether the action of Mg is indirect or not in such degenerative disorders. The study does not describe whether the participants were prescribed to any drug or topical treatment for the OA, or the ingestion of alcohol.
Additionally, Results and Discussion sections could be enlarged including the following issues:
PASE points correlation with the SCREEN score (not included in the study, but a significant measurement for the latter).
BMI values are in the range 25-29.9, that means overweight. Although this condition seems to be not correlated with the outcome of the manuscript, the authors should explain whether overweight is a contributing factor in the results.
Minor comments:
* Table 1 should be 26.5 (4.69) instead of 26.5 (4.69
* Delete lines 206-207.
Author Response
Reviewer 2:
To enrich the introduction I suggest the authors to include an explanation for the link of Mg deficiency and inflammation processes. Hence, the readers may ascertain whether the action of Mg is indirect or not in such degenerative disorders.
R: This is a relevant topic. Therefore, we added this sentence in the Introduction section:
“Animal studies, in fact, reported that Mg deficiency induces a relevant inflammatory response finally resulting in leukocyte and macrophage activation, increase in inflammatory cytokines and acute-phase proteins, and excessive production of free radicals.[14]”
The study does not describe whether the participants were prescribed to any drug or topical treatment for the OA, or the ingestion of alcohol.
R: Nice comment. We have now added this information in the Table 1, as you suggested. Both these variables (alcohol intake and treatments of OA), however, did not enter in the fully-adjusted model since these parameters did not satisfy the criteria for entering among the covariates used.
Additionally, Results and Discussion sections could be enlarged including the following issues:
PASE points correlation with the SCREEN score (not included in the study, but a significant measurement for the latter).
R: Added this important issue in the shortcomings of the work.
BMI values are in the range 25-29.9, that means overweight. Although this condition seems to be not correlated with the outcome of the manuscript, the authors should explain whether overweight is a contributing factor in the results.
R: We sincerely thank the Reviewer for this comment. We have added in the Discussion section the following sentence to support this actual issue:
“In this regard, to be overweight or obese can contribute to knee OA by increasing pressure on lower limbs joints. Excessive weight, in fact, can increase the stress on joints and finally lead to the cartilage damage.[28] Therefore, a higher dietary intake of Mg could delay the onset or attenuate the initial phase of OA in people at higher risk, also because Mg intake seems to be associated with a better body composition in older people.[7]”
Round 2
Reviewer 1 Report
Suggested modifications improved the quality & presentation.
Reviewer 2 Report
The authors have improved the manuscript following the suggestion
This manuscript is a resubmission of an earlier submission. The following is a list of the peer review reports and author responses from that submission.
Round 1
Reviewer 1 Report
The author must show some MR images for the different groups.
Discussion and conclusion need to improve a lot. Discussion should explain more about the data as well as the outcome clearly. Also needs to add more references in the discussion part.
Reviewer 2 Report
The relationship between magnesium intake / serum magnesium and knee osteoarthritis has been studied extensively in the literature. I don't see how much new information has been added from this study. Methodologically, it is not clear why Mg wasn't assessed using quartile comparison to adopt a residual method to adjust for energy intake. It is not clear what are the food sources for Mg in this cohort and how it is related to the food consumption in a Mediterranean diet.